# The Cytomegalovirus Tegument Protein UL35 Antagonizes Pattern Recognition Receptor-Mediated Type I IFN Transcription

**DOI:** 10.3390/microorganisms8060790

**Published:** 2020-05-26

**Authors:** Markus Fabits, Vladimir Gonçalves Magalhães, Baca Chan, Virginie Girault, Endrit Elbasani, Elisa Rossetti, Eirikur Saeland, Martin Messerle, Andreas Pichlmair, Vanda Juranić Lisnić, Melanie M. Brinkmann

**Affiliations:** 1Institute of Genetics, Technische Universität Braunschweig, 38106 Braunschweig, Germany; markus.fabits@helmholtz-hzi.de; 2Viral Immune Modulation Research Group, Helmholtz Centre for Infection Research, 38124 Braunschweig, Germany; v.magalhaes@dkfz-heidelberg.de (V.G.M.); baca.chan@uwa.edu.au (B.C.); 3Institute of Virology, Technical University of Munich, School of Medicine, 81675 Munich, Germany; v.girault@tum.de (V.G.); andreas.pichlmair@tum.de (A.P.); 4Institute of Virology, Hannover Medical School, 30625 Hannover, Germany; endrit.elbasani@helsinki.fi (E.E.); messerle.martin@mh-hannover.de (M.M.); 5Janssen Vaccines & Prevention BV, 2333 CN Leiden, The Netherlands; erosset1@its.jnj.com (E.R.); esaeland@its.jnj.com (E.S.); 6Department for Histology and Embryology, Faculty of Medicine, University of Rijeka, 51000 Rijeka, Croatia; vanda.juranic@uniri.hr

**Keywords:** herpesvirus, cytomegalovirus, pattern recognition receptor, cGAS, STING, RIG-I, TBK1, UL35, type I interferon, OGT

## Abstract

The rapid activation of pattern recognition receptor (PRR)-mediated type I interferon (IFN) signaling is crucial for the host response to infection. In turn, human cytomegalovirus (HCMV) must evade this potent response to establish life-long infection. Here, we reveal that the HCMV tegument protein UL35 antagonizes the activation of type I IFN transcription downstream of the DNA and RNA sensors cGAS and RIG-I, respectively. We show that ectopic expression of UL35 diminishes the type I IFN response, while infection with a recombinant HCMV lacking UL35 induces an elevated type I IFN response compared to wildtype HCMV. With a series of luciferase reporter assays and the analysis of signaling kinetics upon HCMV infection, we observed that UL35 downmodulates PRR signaling at the level of the key signaling factor TANK-binding kinase 1 (TBK1). Finally, we demonstrate that UL35 and TBK1 co-immunoprecipitate when co-expressed in HEK293T cells. In addition, we show that a previously reported cellular binding partner of UL35, O-GlcNAc transferase (OGT), post-translationally GlcNAcylates UL35, but that this modification is not required for the antagonizing effect of UL35 on PRR signaling. In summary, we have identified UL35 as the first HCMV protein to antagonize the type I IFN response at the level of TBK1, thereby enriching our understanding of how this important herpesvirus escapes host immune responses.

## 1. Introduction

Human cytomegalovirus (HCMV) is an opportunistic pathogen that is distributed worldwide with seroprevalences between 45% and 100% depending on location, age, gender, and socioeconomic status [1]. As characteristic for other herpesviruses, HCMV persists life-long in a dormant state and can sporadically re-enter the lytic replication cycle. While lytic infection of healthy individuals is mostly asymptomatic, immunocompromised individuals (e.g., transplant recipients and HIV/AIDS patients) can develop life-threatening disseminating disease [2]. Moreover, HCMV congenital infections are the underappreciated leading cause of congenital birth defects [3]. The immediate recognition of invading pathogens is essential for the induction of a potent host immune response, leading to eventual control of the infection. In order to establish life-long infections, herpesviruses have evolved sophisticated strategies to modulate hosts’ immune responses [4].

Cellular pattern recognition receptors (PRR) sense pathogen-associated molecular patterns (PAMP) such as nucleic acids and immediately activate the transcription of cytokine and antiviral factor genes [5]. Cytosolic DNA can be detected by cyclic guanosine monophosphate-adenosine monophosphate synthase (cGAS) [6] as well as other cytosolic DNA sensors. DNA binding to cGAS facilitates the generation of the second messenger cyclic guanosine monophosphate-adenosine monophosphate (cGAMP) which activates the endoplasmic reticulum (ER) resident protein stimulator of interferon genes (STING) [7]. STING undergoes dimerization and translocates to the Golgi compartment where it forms a platform for the activation of downstream effector proteins including TANK-binding kinase 1 (TBK1) and the transcription factor interferon regulatory factor 3 (IRF3) [8]. TBK1 gets recruited, autophosphorylates itself and subsequently phosphorylates STING [9]. STING phosphorylation enables IRF3 binding and TBK1-mediated IRF3 phosphorylation triggers IRF3 dimerization and its nuclear translocation, from where it facilitates IFNβ transcription. Similarly, RNA can be detected by cytosolic RIG-I-like receptors (RLR) such as retinoic acid inducible gene I (RIG-I) or melanoma differentiation-associated protein 5 (MDA5) [10,11]. Following ligand binding, the adaptor protein mitochondrial antiviral-signaling protein (MAVS) multimerizes and forms a scaffold for TBK1 and IRF3, leading to the phosphorylation and nuclear translocation of IRF3 and subsequent IFNβ transcription [8]. IFNβ protein produced following PRR-mediated activation is then secreted and binds to the type I IFN receptor (IFNAR) in an autocrine and paracrine manner. Activation of IFNAR signaling triggers the JAK-STAT pathway leading to the transcription of multiple interferon-stimulated genes (ISG), a hallmark of the antiviral state [12].

Herpesviruses have developed numerous strategies to evade or impede the host immune response including PRR-signaling and downstream effectors [13,14]. Recently, we identified the murine CMV (MCMV) M35 protein as a type I IFN antagonist that downmodulates type I IFN transcription downstream of multiple PRR [15]. The M35 protein is present in the tegument of the MCMV virion and can therefore act immediately after viral entry into the host cell, without prior requirement for viral gene expression.

In this study, we investigated whether the HCMV homologue of M35, UL35, can likewise modulate PRR signaling. Although M35 and UL35 share only 25% amino acid identity, structure remodeling using the human herpesvirus 6B (HHV-6B) homologue U14 as the template [16] predicted a high similarity for their three-dimensional protein structure. The UL35 open reading frame (ORF) codes for the 75 kDa ppUL35 phosphoprotein (herein referred to as UL35) which was identified as a minor tegument component of HCMV [17]. In addition to UL35, the UL35 locus encodes a second protein, ppUL35a (herein referred to as UL35a), which is not present in the tegument [17]. UL35a is identical to the C-terminus of UL35 as it is translated from an internal start codon leading to a 21 kDa protein. While tegument UL35 is expressed late, UL35a shows early and late expression kinetics during infection [17]. It has been reported that deletion of the complete UL35 ORF led to a strong defect in virus replication, assembly, and particle formation, depending on the multiplicity of infection (MOI) [18]. A recent study showed that loss of UL35a led to a comparable growth deficit as the complete deletion of the UL35 ORF [19]. In contrast, when only UL35 expression was impaired but expression of UL35a was unaffected, a moderate defect in viral replication was observed [19]. UL35 was shown to positively affect the activation of the viral major immediate-early promoter (MIEP) by interacting with the viral transactivator pp71, encoded by UL82 [20].

Another example of the complex interplay between UL35, UL35a, and UL82 was reported by Salsman and colleagues in 2011 [21]. Overexpression experiments revealed that UL35 forms ring-like nuclear bodies independent of the promyelocytic leukemia (PML) protein and co-localizes with PML and PML-associated proteins such as Daxx and SP100 [21]. Additionally, the number of nuclear bodies formed by UL35 was enhanced by co-expressed UL82. When expressed individually, UL82 and UL35a localize to the nucleus, while they localize to the cytoplasm upon co-expression [21]. These divergent functions of UL35 and UL35a underline the importance of their tight temporal control of their expression through the infection cycle. In 2012, Salsman et al. revealed multiple cellular interaction partners of transiently expressed UL35 and UL35a [22]. DCAF1, a member of the Cul4^DCAF1^ E3 ubiquitin ligase complex, was identified as an interaction partner of UL35, but not UL35a. DCAF1 co-localizes with the ring-like UL35 nuclear bodies [22].

Here, we identify the HCMV tegument protein UL35 as a novel type I IFN antagonist which acts downstream of the PRR cGAS and RIG-I, but upstream of the IFNAR signaling pathway. Upon HCMV infection, tegument UL35 antagonizes the proper activation of the STING-TBK1-IRF3 signaling platform, leading to impaired activation of IFNβ transcription and ISG expression.

## 2. Materials and Methods

### 2.1. Cell Lines

Human embryonic kidney 293T/17 cells (HEK293T, CRL-11268™) were obtained from American Type Culture Collection, Manassas, VA, USA (ATCC) and maintained in Dulbecco’s modified Eagle’s medium (DMEM; high glucose, Sigma-Aldrich, Darmstadt, Germany) supplemented with 8% fetal calf serum (FCS) and 1× penicillin/streptomycin (P/S), Gibco (Sigma-Aldrich). Primary human foreskin fibroblasts, HFF-1, were obtained from ATCC (SCRC-1041™) and cultivated in DMEM high glucose supplemented with 10% FCS, 1 mM sodium pyruvate, and 1x P/S. Human lung fibroblasts MRC-5 were obtained from ATCC (CCL-171™) and maintained in DMEM, high glucose, supplemented with 10% FCS and 1× P/S. Cells were grown in a humidified incubator at 37 °C with 7 Vol% CO_2_. HFF-1 stably expressing empty vector (ev) or UL35-HA were generated by retroviral transduction using pMSCVpuro expression constructs and selection with 2 µg/mL puromycin.

### 2.2. Antibodies and Reagents

Rabbit anti-HA (#3724 clone C29F4), rabbit anti-IRF3 (#11904 clone D6I4C), rabbit anti-p65 (#6956 clone L8F6), rabbit anti-Fibrillarin (#2639 clone C13C3), mouse anti-myc (#2276 clone 9B11), rabbit anti-OGT (#24083 clone D1D8Q), mouse anti-O-GlcNAc (#9875 clone CTD110.6), rabbit anti-TBK1 (#3504 clone D1B4), rabbit anti-p-TBK1 (#5483 clone D52C2), rabbit anti-p-IRF3 Ser396 (#29047 clone D6O1M), rabbit anti-p-STING Ser366 (#85735), and rabbit anti-p-STAT1 Tyr701 (#7649 clone D4A7) were purchased from Cell Signaling Technology (Frankfurt am Main, Germany). Mouse anti-α-Tubulin (#T6199) and mouse anti-β Actin (A5441) were obtained from Sigma-Aldrich. Rat anti-HA-HRP (#12013819001 clone 3F10) was purchased from Roche (Mannheim, Germany). Mouse anti-pp65 (#ab6503 clone 3A12) was purchased from Abcam (Cambridge, UK) and mouse anti-UL44 ICP36 (#MBS530793) was obtained from MyBioSource (San Diego, CA, USA). Mouse anti-IE1 (clone 63-27), originally described in [23], was a kind gift from Jens von Einem (Institute of Virology, Ulm University Medical Center, Ulm). Rabbit anti-UL82 (clone SA2932), originally described in [24], was provided by Thomas Stamminger (Institute of Virology, Ulm University Medical Center, Ulm). HRP-conjugated and Alexa Fluor^®^-conjugated secondary antibodies were purchased from Dianova (Hamburg, Germany) and Invitrogen (Life-Technologies, Darmstadt, Germany), respectively.

For production of mouse monoclonal anti-UL35 antibody, a section of UL35 corresponding to aa 465–641 of the UL35 ORF was subcloned into pET-28c (Novagen, Merck Millipore, Darmstadt, Germany). Recombinant protein was expressed in *E. coli* strain BL21 (DE3) via IPTG induction. Immunization of mice and generation of hybridoma cultures was performed as reported previously [25]. Specificity of antibodies was validated by ELISA on UL35 peptide used for immunization versus irrelevant His-tagged peptide. Antibodies were further tested on UL35 expressing cell lysates by immunoblotting, immunoprecipitation and immunofluorescence. Selected antibodies were purified from hybridoma supernatants using protein G columns (GE Healthcare, Chicago, IL, USA) on Äkta Prime Plus.

High molecular weight poly(I:C) was purchased from Invivogen (San Diego, CA, USA) (#tlrl-pic). Interferon-stimulatory DNA (ISD) was generated by the combination of complementary forward (ISD45 bp-for: 5′-TACAGATCTACTAGTGATCTATGACTGATCTGTACATGATCTACA) and reverse (ISD45 bp-rev: 5′-TGTAGATCATGTACAGATCAGTCATAGATCACTAGTAGATCTGTA) 45 bp oligonucleotides, heating to 70 °C for 10 min followed by annealing at room temperature. Protease inhibitors (4693116001) and phosphatase inhibitors (4906837001) were purchased from Roche. For transfections, Lipofectamine 2000, FuGENE HD, and polyethylenimine (PEI) were purchased from Life-Technologies, Promega (Walldorf, Germany), and Polysciences, Inc. (Warrington, PA, USA), respectively. JetPEI was obtained from Polyplus transfection (Illkirch, France) and OptiMEM was obtained from Thermo Fisher Scientific (Darmstadt, Germany). Recombinant human IFNβ (#300-02BC) was ordered from PeproTech (Hamburg, Germany).

### 2.3. Plasmids

Expression constructs of HA-tagged and untagged UL35 were generated by subcloning PCR amplified ORF UL35 (GenBank accession# AAR31600.1) into pcDNA3.1+ (Invitrogen) via HindIII/NotI sites:

(HindIII-UL35_for 5′-GCATAAGCTTGCCACCATGGCCCAGGGATCGCGAGC-3′ and NotI-UL35-untagged_rev 5′-CCATGCGGCCGCtcaGAGATGCCGTAGGTTTTCGGC-3′ or NotI-UL35-HA_rev 5′-CCATGCGGCCGCctaTGCGTAGTCTGGTACGTCGTACGGATATGCGTAGTCTGGTACGTCGTACGGATAGAGATGCCGTAGGTTTTCG-3′). HA-tagged UL35 was subcloned into pMSCVpuro (Clontech) via blunt end cloning using HpaI/PmeI sites to generate pMSCVpuro UL35-HA. pEFBOS mCherry-mSTING (designated Cherry-STING) expressing N-terminal monomeric Cherry fused to murine STING and pIRESneo3 cGAS-GFP (GFP fused to the C-terminus of human cGAS) were kindly provided by Andrea Ablasser (Global Health Institute, Ecole Polytechnique Fédérale de Lausanne, Switzerland). The Renilla luciferase expression construct pRL-TK and pIRES2-GFP were purchased from Promega and Clontech, respectively. pGL3basic IFNβ-Luc (IFNβ-Luc) and pGL3basic ISG56-Luc (ISG56-Luc) were described previously [15]. pcDNA3-FLAG-TBK1 was described previously by Søren Paludan, Aarhus University, Denmark [26]. CMVBL IRF3-5D codes for human IRF3 containing five amino acid substitutions (S396D, S398D, S402D, S404D, S405D) which renders it constitutively active and was provided by John Hiscott (Institut Pasteur Cenci Bolognetti Foundation, Rome, Italy). pCAGGS Flag-RIG-I N, expressing a constitutively active truncation mutant of RIG-I, was kindly provided by Andreas Pichlmair (Technical University Munich, Germany). pFLAG-CMV2-MAVS was described previously [27] and was kindly provided by Friedemann Weber (Justus-Liebig-Universität Giessen, Germany). pcDNA4/LacZ-myc/His was purchased from Invitrogen. C-terminally myc/His tagged M76 was subcloned from the Smith strain MCMV BAC into pcDNA4B myc/His (Invitrogen) using HindIII/XbaI sites. pcDNA4-M35-myc/His was previously described [15]. Expression constructs for pcDNA3.1 M35-V5/His and M27-V5/His have been described previously [28].

O-GlcNAcylation mutants of UL35 (all C-terminally HA tagged, pcDNA3.1+) were generated using the Q5 site-directed mutagenesis kit (New England Biolabs, Frankfurt am Main, Germany #E0554) according to the manufacturer’s instructions. For UL35 Ala529-553, serines and threonines within UL35 aa 529–553 were substituted for alanine. For UL35 Ala529-531, threonines at aa position 529–531 were substituted for alanines. For UL35 Ala534/537, serine 534 and threonine 537 were mutated to alanines. UL35 Ala550-553 was generated by mutating serines at position 550–553 to alanines. The expression construct of untagged OGT was subcloned from pOTB7-OGT (accession #BC014434, cDNA obtained from Dharmacon, Lafayette, CO, USA) into pcDNA3.1+ via BamHI/NotI sites.

Primer sequences as well as sequences of all constructs are available upon request.

### 2.4. BAC Mutagenesis

Manipulation of the UL35 open reading frame was carried out by *en passant* mutagenesis [29] on the HCMV bacterial artificial chromosome (BAC) TB40/E-BAC4 (GenBank accession # EF999921.1) [30]. To introduce a stop cassette (GGCTAGTAATAGCCT) at nucleotide position 228 within the UL35 ORF (nucleotide position 79145 of the HCMV genome), a linear PCR construct was generated using pori6K-RIT [31] as template and the primers UL35stopEPfor: 5′-GGAGGCCCTGGTGGACTTCCAGGTGCGCAACGCTTTTATG**GGCTAGTAATAGCCT**AAGGTAAAGCCCGTGGCCCAgacgcatcgtggccggatctc-3′ and UL35stopEPrev: 5′-TGCAGATACGGATAATCTCCTGGGCCACGGGCTTTACCTT**AGGCTATTACTAGCC**CATAAAAGCGTTGCGCACCTgtgaccacgtcgtggaatgc-3′. HCMV specific sequences are underlined and the stop cassette is shown in bold. The PCR product was purified and transformed in GS1783 bacteria harboring TB40/E-BAC4 [30]. Afterwards, the two-step recombination process was carried out essentially as described in [29]. The resultant UL35stop HCMV BAC was fully sequenced by paired-end sequencing using the MiSeq system from Illumina.

### 2.5. HCMV Reconstitution

For virus reconstitution, endotoxin-free HCMV BAC DNA (TB40/E-BAC4 wildtype and UL35stop) was isolated from *E. coli* strain GS1783 using the NucleoBond^®^ Xtra Midi EF kit (Macherey-Nagel, Düren, Germany). Isolated BAC DNA was transfected into MRC-5 cells using JetPEI (Polyplus transfection, Illkirch, France) and cells were cultivated until cytopathic effect was visible in the majority of cells. Infectious supernatant was used to infect HFF-1 seeded in T25 flasks. Approximately 10 days post infection, infectious HCMV containing supernatant was diluted with fresh HFF-1 medium and used to infect two T175 flasks containing 4 × 10^7^ HFF-1 cells. Approximately 12 days later, supernatant and cells were removed and centrifuged at 2465× *g* to remove cell debris. Supernatant was further subjected to centrifugation at 26,000× *g* for 3 h at 4 °C to pellet virus. Subsequently, virus particles were purified on a Nycodenz cushion by centrifugation at 46,000× *g* for 3 h at 4 °C. Purified, pelleted virus was resuspended in VSB buffer (50 mM Tris-HCl, pH 7.8, 12 mM KCl, 5 mM Na_2_EDTA), aliquoted and stored at −70 °C.

### 2.6. HCMV Titration

HCMV titers were determined by standard plaque assay on HFF-1 cells. In short, HFF-1 seeded in 48-well plates were infected with serial dilutions of HCMV virus and overlaid with carboxymethylcellulose (CMC)-DMEM medium (DMEM supplemented with 11.64 g/L CMC, 5% FCS, 2 mM glutamine, 1× P/S, 0.37% NaHCO_3_ and 0.351% D-(+)-glucose). Plaque formation was monitored and counted approximately 14 days post infection. Titers were expressed as PFU/mL and used to calculate the MOI for infection experiments.

### 2.7. Immunolabeling of HCMV IE1 Antigen

IE1 immunolabeling was routinely performed to ensure equal input of HCMV WT and UL35stop in infection experiments and to determine HCMV titers from growth curve experiments. HFF-1 cells (5,000 cells per well of a 96-well plate) were infected with serial dilutions of HCMV virus stocks or supernatant from infected cells. 48 h post infection, cells were washed with PBS and fixed with 4% paraformaldehyde (PFA) for 10 min. Fixed cells were permeabilized with 0.1% Triton X-100 for 5 min and blocked with PBS containing 5% FCS and 1% BSA for 30 min. Next, cells were immunolabeled with mouse anti-IE1 antibody, washed with PBS and subsequently labeled with secondary anti-mouse AF488 conjugated antibody. IE1 positive nuclei were counted using an IncuCyte S3 (Essen BioSciences, Sartorius, Göttingen, Germany) and resulting titers were calculated and expressed as IE1^+^ cells/mL.

### 2.8. Growth Curve

HFF-1 cells were seeded onto 48-well plates (35,000 cells per well) and infected with HCMV WT or UL35stop at an MOI of 0.1. At 0, 3, 5, 7, 9, 11, or 13 days post infection, supernatants were harvested and freshly seeded HFF-1 were infected with serial dilutions of the harvested supernatants to perform IE1 immunolabeling. 48 h post infection, cells were immunolabeled for IE1 as described above to calculate HCMV titers at the individual time points.

### 2.9. Luciferase-Based Reporter Assays

For all reporter assays, 25,000 HEK293T cells were seeded per well in 96-well plates.

cGAS-STING assay: HEK293T cells were transiently transfected with 120 ng plasmid of interest or empty vector (pcDNA3.1+), 60 ng of pEFBOS-cGAS (stimulated) or pIRES2-GFP (unstimulated), 60 ng pEFBOS-mCherry-STING, 100 ng pGL3basic IFNβ-Luc, 10 ng pRL-TK and 1.2 μL of Fugene HD (Promega) diluted in a total volume of 10 µL Opti-MEM (Thermo Fisher Scientific).

MAVS assay: HEK293T cells were transiently transfected with 5 ng pFLAG-CMV2-MAVS (stimulated) or pcDNA3.1+ (unstimulated) together with 100 ng pGL3basic IFNβ-Luc, 10 ng pRL-TK, and 50 ng plasmid of interest complexed with 0.6 μL FuGENE HD in 10 µL Opti-MEM.

RIG-I N: HEK293T cells were transiently transfected with 13 ng pCAGGS Flag-RIG-I N (stimulated) or pcDNA3.1+ (unstimulated) together with 50 ng pGL3basic IFNβ-Luc, 5 ng pRL-TK, and 130 ng plasmid of interest complexed with 0.66 μL FuGENE HD in 10 µL Opti-MEM.

TBK1 assay: HEK293T cells were transiently transfected with 100 ng pcDNA3-FLAG-TBK1 or 100 ng pIRES2-GFP (unstimulated), 100 ng pGL3basic IFNβ-Luc, 10 ng pRL-TK, 120 ng plasmid of interest, and 1 μL FuGENE HD diluted in 10 µL OptiMEM.

IRF3-5D: HEK293T cells were transiently transfected with 120 ng plasmid of interest, 60 ng of IRF3-5D (stimulated) or pIRES2-GFP (unstimulated), 100 ng pGL3basic-IFNβ-Luc, 10 ng pRL-TK and 1 μL of Fugene HD diluted in 10 µL Opti-MEM.

Cells from cGAS-STING, RIG-I N, MAVS, TBK1 and IRF3-5D reporter assays were lysed in 1 × passive lysis buffer (PLB) (Promega) 20 h post transfection.

IFNβ/ISG56 assay: HEK293T cells were transiently transfected with 120 ng plasmid of interest or pcDNA3.1+, 100 ng pGL3basic-ISG56-Luc, 10 ng pRL-TK and 0.8 μL of Fugene HD diluted in Opti-MEM. 24 h post transfection, cells were stimulated with 0.1 ng/mL recombinant human IFNβ (PeproTech, #300-02BC) or mock stimulated and lysed 16 h later in 1 × PLB.

Luciferase production was measured with the dual-luciferase reporter assay system (Promega) and the Tecan Infinite^®^ 200 Pro microplate luminometer (Tecan, Männedorf, Switzerland). Luciferase fold induction was calculated by dividing Renilla-normalized values from stimulated samples by the corresponding values from unstimulated samples.

### 2.10. siRNA Knockdown

Briefly, 15,000 HEK293T cells were combined with 50 nM siRNA complexed with 0.3 µL Lipofectamine 2000 and simultaneously seeded into 96-well plates. 48 h later, cells were transfected for luciferase reporter assays as described and luciferase activity was measured 20 h later. The following siRNAs were obtained from Dharmacon: ON-target plus non-targeting pool (#D-001810-10-05), SMARTpool siGENOME human OGT (#M-019111-00-0005), SMARTpool siGENOME human DCAF1 (#M-021119-01-0005), SMARTpool siGENOME human CUL4A (#M-012610-01-0005).

### 2.11. Immunoblotting

For the immunoblotting of luciferase samples, 20 µL of lysates (prepared in 1 × PLB) were boiled for 10 min at 95 °C in SDS sample buffer and subjected to SDS-PAGE. For immunoblotting of HCMV viral particles, 5 × 10^4^ PFU from purified virus preparations were heated in SDS sample buffer and separated by SDS-PAGE. For analysis of HCMV protein expression levels, 100,000 HFF-1 (seeded in 24-well plates) were infected with either HCMV WT or UL35stop at an MOI of 0.1. Infection was enhanced by centrifugation at 800× *g* for 45 min at room temperature (RT). Afterwards, cells were washed with HFF-1 medium and incubated for the indicated time points until lysis. Whole cell lysates were prepared using radioimmunoprecipitation assay (RIPA) buffer (20 mM Tris-HCl pH 7.5, 1 mM EDTA, 100 mM NaCl, 1% Triton X-100, 0.5% sodium deoxycholate, 0.1% SDS) supplemented with protease and phosphatase inhibitors. For kinetic analysis of phosphorylated proteins, HFF-1 pMSCV or pMSCV UL35 were seeded into 24-well plates (100,000 cells per well) and were mock treated or infected with HCMV UL35stop at an MOI of 0.1 with 15 min enhancement via centrifugation (700× *g*, RT). Cells were further incubated for 30 min before the supernatant was exchanged with fresh HFF-1 medium. Cells were lysed at indicated time points in lysis buffer containing 25 mM Tris-HCl pH 7.6, 150 mM NaCl, 1% IGEPAL CA-630, 1% sodium deoxycholate, 0.1% SDS, protease inhibitors and phosphatase inhibitors. For ISD stimulation, cells were seeded as described for UL35stop infection and either Lipofectamine 2000 containing OptiMEM was added (unstimulated control) or 10 µg/mL ISD complexed with Lipofectamine 2000 in OptiMEM. Cells were lysed at indicated time points.

Cell lysates were cleared by centrifugation at 17,000× g following separation by SDS-PAGE and transfer onto nitrocellulose or PVDF membranes (both GE Healthcare). Primary antibodies were added as indicated followed by incubation with secondary horseradish peroxidase (HRP) coupled antibodies. Membranes were developed with Lumi-Light (Roche Applied Science) or SuperSignal West Femto (Thermo Scientific) chemiluminescence substrates and imaged on a ChemoStar ECL Imager (INTAS, Göttingen, Germany).

For co-immunoprecipitation (coIP) experiments, 850,000 HEK293T cells were transiently transfected with 3 µg total DNA complexed with 12 µL PEI diluted in a total volume of 200 µL PBS. 24 h post transfection, cells were lysed in IP lysis buffer (20 mM Tris-HCl pH 7.5, 150 mM NaCl, 1% IGEPAL CA-630 (NP-40 replacement), 0.25% sodium deoxycholate, 0.1% SDS) freshly supplemented with protease inhibitors for 3 h at 4 °C on a rotator. 10% of the lysate was used as input control, and the remaining IP fraction was pre-cleared with protein A agarose beads (IPA300, Repligen) for 1 h. Cleared lysates were then incubated overnight with respective antibodies at 4 °C before addition of protein A agarose beads for 1 h. Beads were washed 7 times with IP lysis buffer and bound protein was eluted by heating samples in SDS sample buffer. Input lysates and IP samples were then analyzed by immunoblotting.

### 2.12. Subcellular Fractionation

HFF-1 were seeded at a density of 200,000 cells into a 12-well plate. The next day, cells were infected with HCMV WT or UL35stop at an MOI of 0.5 by centrifugal enhancement for 45 min at 800× *g* and RT. After centrifugation, cells were washed with HFF-1 medium and incubated until fractionation at 37 °C and 7% CO_2_. At the indicated time points, cells were collected in tubes containing 300 μL cold PBS and centrifuged at 10,416× *g* for 10 s at 4 °C. Pellets were lysed with 300 μL ice-cold 0.1% IGEPAL CA-630 in PBS. 45 μL of the lysate were removed and combined with 15 μL of 4× SDS sample buffer and designated as the whole cell lysate (WCL). The remaining lysate was centrifuged at 16,873× *g* for 10 s at RT and 45 μL of supernatant were added to 15 μL 4× SDS sample buffer and designated as the cytosolic fraction (C). The pellet was washed with 300 μL 0.1% IGEPAL CA-630 in PBS followed by centrifugation at 16,873× *g* for 10 s at 4 °C. The pellet was resuspended in 30 μL of 1× SDS sample buffer and designated as the nuclear fraction (N). WCL and N fractions were sonicated using a Bioraptor device at high power settings (30 sec on/30 sec off, 6 cycles). For immunoblotting, 10 μL of the WCL and C fractions and 5 μL of the N fraction were loaded per lane on SDS gel.

### 2.13. Proteomic Analysis

For proteomic analysis, 1 × 10^6^ HFF-1 stably expressing pMSCV or pMSCV UL35 were left untreated or stimulated with 10 µg/mL poly(I:C) complexed with Lipofectamine 2000. Four hours later, cells were washed with cold PBS and harvested. Cells were pelleted (300× *g*, 5 min), shock frozen in liquid nitrogen and kept at –80 °C until analysis. All samples were generated in quadruplicates. Cell pellets were thawed on ice for 30 min and 60 µL 10× guanidinium choride buffer (6 M guanidinium chloride, 10 mM TCEP, 40 mM CAA and 0.1 M Tris-HCl pH 8) were added for an additional 30 min. Samples were heated at 99 °C for 15 min with gentle agitation (500 rpm) and sonicated for 5 min at 4 °C and high frequency (Bioruptor). After centrifugation at 15,000× *g* for 30 min at 4 °C, the protein concentration in the supernatant fraction was measured on a Nanodrop™ 1000 spectrophotometer. Subsequently, 50 µg protein were used and resuspended in 20 µL of 0.1 M Tris-HCl pH 8. Proteins were digested into peptides with 1 µg LysC (Wako, Neuss, Germany #129-02541) for 3 h at 37 °C and 0.5 µg Trypsin (Promega # V5113) overnight at 30 °C. Peptides were purified and concentrated on stage tips with three C18 Empore filter discs (3M, Neuss, Germany) and analyzed by mass spectrometry as described before [32]. Briefly, proteomes were measured via LC-MS/MS, using an EASY-nLC 1200 system (Thermo Fisher Scientific) coupled to a LTQ-Orbitrap XL mass spectrometer (Thermo Fisher Scientific). Peptides were loaded on a 20 cm reverse-phase analytical column (75 μm column diameter; ReproSil-Pur C18-AQ 1.9 μm resin; Dr. Maisch, Ammerbuch, Germany) and separated using a 120 min acetonitrile gradient. Raw files were processed using MaxQuant version 1.6.0.15 with label-free quantification (LFQ) and Match between Runs options as described before [32]. Using Perseus version 1.6.0.7., protein groups were filtered for reverse identification, modification site only identification, MaxQuant contaminants list and if not identified in three over four technical replicates in at least one condition. Significant protein expression changes between UL35 expressing cells and their corresponding empty vector control were determined by a two-sided Student’s *t*-test (S_0_ = 0.1) and corrected for multiple hypothesis testing using permutation-based FDR statistics (FDR = 0.05, 250 permutations).

### 2.14. RT-qPCR

For all RT-qPCR experiments, 200,000 HFF-1 cells were seeded per well in 12-well plates the day before infection or stimulation.

ISD stimulation: HFF-1 cells stably expressing pMSCV or pMSCV UL35 were mock treated or transfected with OptiMEM containing 2.5 µg/mL ISD complexed with Lipofectamine 2000. Four hours post transfection, RNA was prepared using the innuPREP RNA mini kit 2.0 (Analytik Jena, Jena, Germany) according to the manufacturer’s instructions.

Sendai virus infection: HFF-1 cells stably expressing pMSCV or pMSCV UL35 were mock treated or infected with OptiMEM containing SeV for 1 h. Afterwards, medium was exchanged, and RNA was extracted 6 h post infection.

HCMV infection: Primary HFF-1 infected with either HCMV WT, HCMV UL35stop, or mock treated. To enhance infection efficiency, cells were centrifuged for 45 min at 800× g. Total RNA was extracted 6 h post infection. In parallel, aliquots of diluted HCMV WT and UL35stop virus used for infection were analyzed by IE1 immunofluorescence to control for equal inoculation doses of WT and UL35stop.

IFNβ stimulation: HFF-1 cells stably expressing pMSCV or pMSCV UL35 were mock treated or stimulated with 20 ng/mL human recombinant IFNβ for 6 h. Medium was removed, and RNA extracted from infected cells.

For all RT-qPCR experiments, genomic DNA was removed using the DNA-free kit (Ambion, Thermo Fisher) and 200 ng total RNA was used to generate cDNA (iScript cDNA synthesis kit-BioRad, Feldkirchen, Germany) according to the manufacturer’s instructions. Quantification of mRNA transcripts was performed using the GoTaq qPCR Master Mix 2× (Promega) on a LightCycler 96 instrument (Roche). qPCR primers were as follows: hHPRT1 for: 5′-GAACGTCTTGCTCGAGATGTG-3′, hHPRT1 rev: 5′-CCAGCAGGTCAGCAAAGAATT-3′, hIFNB1 for: 5′-TGTGGCAATTGAATGGGAGGCTTGA-3′, hIFNB1 rev: 5′-TCAATGCGGCGTCCTCCTTCTG-3′, hIFIT3 for: 5′-CCTACATAAAACACCTAGATGGT-3′, hIFIT3 rev: 5′-AAGTGATAGTAGACCCAGGCGT-3′, hIFIT1 for: 5′-CACCATTGGCTGCTGTTTAG-3′, hIFIT1 rev: 5′-CTCCTCTGAGATCTGGCTATTC-3′. Relative fold inductions were calculated using the 2^−ΔΔCt^ method.

### 2.15. MSD Multiplex Assay

HFF-1 cells seeded into 12-well plates (200,000 cells per well) were infected with HCMV WT or UL35stop at an MOI of 0.5 by centrifugal enhancement (800× *g*, 45 min, RT). Cells were incubated for 2 h at 37 °C, 7 Vol% CO_2_ before medium was exchanged with fresh medium. Supernatants were harvested 24 h and 48 h post infection and used to measure secreted IFNβ using the U-Plex human interferon combo kit (Meso Scale Discovery, MSD, Rockville, MD, USA #K15094K) according to the manufacturer’s protocol. The plates were analyzed on a MESO™ Sector 600 using Discovery workbench 4.0.

### 2.16. Immunofluorescence

HFF-1 stably expressing pMSCV or pMSCV UL35-HA were seeded on acid washed glass cover slips in 24-well plates (40,000 cells per well) one day prior immunolabeling. The next day, medium was removed, and cells were fixed with 4 % PFA in PBS for 20 min at RT. Fixed cells were washed with PBS and remaining PFA was inactivated with 50 mM NH_4_Cl for 10 min. Cells were washed with PBS and permeabilized with 0.4% Triton X-100 for 10 min. After washing with PBS, cells were blocked with 4% BSA for 1 h prior to incubation with primary antibody. For immunolabeling, rabbit anti-HA (C29F4) was added (1:1800) and incubated for 1 h at RT. Labeled cells were washed with PBS prior to addition of secondary anti-rabbit coupled to Alexa Fluor 488 (Invitrogen) and 4′,6-diamidino-2-phenylindole (DAPI). Secondary antibody was incubated for 1 h and finally cells were washed with PBS and subsequently embedded in Prolong Gold (Invitrogen).

For infection: 20,000 HFF-1 cells were seeded on µ-slide plastic chambers (ibidi # 80826, Gräfelfing, Germany) one day before infection. The next day, cells were infected with either HCMV WT or HCMV UL35stop at an MOI of 5 for 90 min. Infectious supernatant was removed, cells were washed with fresh HFF-1 medium and incubated for the remaining time at 37 °C, 7 Vol% CO_2_. At the indicated time points post infection, cells were fixed with PFA and immunolabeled as described above. Primary antibodies against mouse anti-UL35 (1:2 hybridoma supernatant) and rabbit anti-UL82 (1:500) were used for labeling. Secondary anti-mouse AF488 and anti-rabbit AF647 (both obtained from Invitrogen) were combined with DAPI.

Imaging was performed on a Nikon ECLIPSE Ti-E-inverted microscope equipped with a spinning disk device (Perkin Elmer Ultraview). For z-stacks, the integrated piezo drive was used in 5 µm steps. Images including 3D reconstruction were processed using Volocity software (version 6.2.1 and 6.5.1, Perkin Elmer, Hamburg, Germany).

### 2.17. Statistics

Differences between two data sets were evaluated by Student’s *t*-test (unpaired, two-tailed) using GraphPad Prism (GraphPad Software, San Diego, CA). *p* values <0.05 were considered statistically significant. * *p* < 0.05, ** *p* < 0.01, *** *p* < 0.001, **** *p* < 0.0001.

## 3. Results

### 3.1. The HCMV Tegument Protein UL35 Antagonizes the Type I Interferon Response Downstream of DNA and RNA Cytosolic Recognition Receptors, but Upstream of the IFNAR

CMV has evolved multiple strategies to circumvent its detection by PRR, which would otherwise lead to a potent antiviral response mediated by type I IFN and the clearance of infection [13,33,34]. Recently, we identified the MCMV tegument protein M35 as a strong antagonist that shuts down type I IFN signaling downstream of all PRR classes [15]. Based on the positional homology between M35 and UL35 [35,36], albeit with only 25% identity on the amino acid level, and the fact that both proteins are incorporated into the virion, we asked whether UL35 exerts a similar negative effect on type I IFN responses as M35.

We first assessed if UL35 is capable of targeting the cGAS-STING pathway, which is essential for a robust type I IFN response to CMV infection [37]. To monitor the induction of IFNβ transcription, HEK293T cells were transfected with a reporter plasmid composed of the murine IFNβ promoter upstream of the firefly luciferase gene (IFNβ-Luc). Since HEK293T cells do not express endogenous cGAS and STING [38], we transfected the cells with a STING expression construct and induced signaling by co-expression of cGAS. In addition, we co-transfected the cells with expression constructs for untagged UL35, C-terminally HA-tagged UL35 or M35, or the respective empty vector. Consistent with our published findings [15], the MCMV M35 protein inhibited cGAS-STING-mediated signaling, and HCMV UL35 also significantly reduced IFNβ transcription compared to ev control (Figure 1A). Next, we wanted to address whether UL35 also inhibits signaling downstream of other PRR. To test this, we expressed a constitutive active mutant of the cytoplasmic RNA sensor RIG-I, RIG-I N. UL35, like M35, clearly inhibited RIG-I signaling (Figure 1B). To determine whether UL35 targets at the level of or downstream of RIG-I, we overexpressed the RIG-I adaptor protein MAVS and observed that UL35 and M35 still inhibited IFNβ transcription (Figure 1C). This suggests that UL35 inhibits signaling at a level where the cGAS-STING and RIG-I-MAVS pathways converge. Next, we overexpressed the downstream kinase TBK1 and observed that UL35 inhibited TBK1-mediated activation of IFNβ transcription to a similar extent as M35 (Figure 1D). Intriguingly, UL35, in contrast to M35, did not inhibit activation of IFNβ expression by a constitutively active form of the transcription factor IRF3, IRF3-5D (Figure 1E). Next, to exclude an effect of UL35 on IFNAR signaling, HEK293T cells were co-transfected with an ISG56 promoter firefly luciferase reporter and IFNAR signaling was stimulated by the addition of exogenous IFNβ. MCMV M27, a known modulator of IFNAR signaling [39], inhibited ISG56 promoter induction, whereas UL35 and M35 did not downmodulate the ISG56 reporter (Figure 1F). Taken together, these data show that UL35 negatively modulates induction of IFNβ transcription downstream of the PRRs cGAS and RIG-I, but leaves IFNAR signaling intact. Since UL35 did not affect IRF3-5D-mediated IFNβ transcription, we conclude that UL35 inhibits the PRR signaling pathway either at the level of TBK1 or between TBK1 and IRF3. This distinguishes it from the MCMV homologue M35, which targets IFNβ transcription from the nucleus.

### 3.2. Generation of a UL35-Deficient Recombinant HCMV for Functional Studies

In order to assess whether UL35 antagonizes the IFNβ response during HCMV infection, we created a UL35-deficient recombinant HCMV by *en passant* mutagenesis, designated HCMV-UL35stop. For this, we introduced a stop cassette at nucleotide position 228 of the UL35 ORF (or nucleotide position 79145 of the HCMV genome strain TB40/E, GenBank accession #EF999921.1) [30] to prematurely terminate UL35 translation (Figure 2A). With this strategy, expression of UL35a is not affected. After virus reconstitution, we subjected HCMV UL35stop virions to immunoblotting with a monoclonal antibody specific for the C-terminus of UL35 and confirmed the absence of UL35 in this recombinant virus (Figure 2B). As a loading control, we verified for the presence of the major tegument protein pp65 (encoded by UL83) which was present at equal levels in HCMV WT and UL35stop virions (Figure 2B).

To assess the impact of the loss of UL35 on HCMV replication, we infected primary human foreskin fibroblasts (HFF-1) with either HCMV WT or UL35stop at an MOI of 0.1 and analyzed the viral titer at the indicated time points. Under these conditions, the viral growth curves of WT and UL35stop were comparable (Figure 2C). Next, we characterized the kinetics of HCMV protein expression in HCMV WT or UL35stop infected primary HFF-1. We prepared whole cell lysates at the indicated time points post infection representing one full cycle of HCMV replication, and analyzed expression of tegument proteins pp65 (UL83), UL35, and the early-late proteins UL44 and UL35a. As expected, we did not detect UL35 in UL35stop-infected cells, whereas UL35 tegument protein was visible at 4, 6, and 10 h post infection with HCMV WT, and de novo expressed UL35 at 48, 72, and 96 h post infection (Figure 2D), as observed by others [17]. The UL35a, UL44 and pp65 (UL83) proteins were expressed with similar and expected kinetics in HCMV WT and UL35stop infected cells (Figure 2D). Overall, we successfully created a recombinant UL35stop HCMV to study the influence of the tegument UL35 protein on PRR signaling during infection.

### 3.3. Tegument UL35 Immediately Translocates to the Nucleus Upon Infection

To determine the subcellular localization of the incoming UL35 protein upon infection, we fractionated lysates of uninfected (mock), HCMV WT, and UL35stop infected HFF-1 and immunoblotted for UL35 with the UL35-specific monoclonal antibody. As early as 1.5 h post infection, UL35 protein was present in the nuclear as well as the cytoplasmic fraction of HCMV WT infected cells (Figure 3A, upper panel). UL35 could still be detected in the cytoplasmic and nuclear fraction at 5 h post infection (Figure 3A, lower panel).

To verify the results of the cellular fractionation study, we analyzed the localization of UL35 by immunofluorescence during infection and included HCMV UL35stop as a negative control. To assess successful infection of UL35stop-infected cells, we labeled for the tegument protein pp71 (encoded by UL82) which is a known binding partner of UL35 [20]. At 2 h post infection, we observed that UL35 and UL82 co-localized in the cytoplasm and in the nuclei of infected cells (Figure 3B). Image 3D reconstruction of z-stacks confirmed that a portion of UL35 accumulated inside the nucleus from 4 h post infection (Figure 3B). Notably, even at 6 h post infection, UL35 was still present in the cytoplasm (Figure 3B), which is consistent with the subcellular fractionation analysis (Figure 3A). These results show the rapid dynamics of incoming tegument UL35 upon HCMV infection, with fast accumulation in the nucleus and continuous presence in the cytoplasm.

### 3.4. Tegument UL35 Negatively Modulates IFNβ Induction in HCMV-Infected Fibroblasts

To verify our findings about the antagonistic activity of UL35 during infection, we analyzed IFNβ transcript levels in HFF-1 upon infection with HCMV WT and UL35stop. At 6 h post infection, IFNβ transcript levels were higher in cells infected with HCMV UL35stop than in cells infected with HCMV WT (Figure 4A). In addition, we assessed IFNβ secretion following HCMV infection of HFF-1 cells by MSD Multiplex assay at 24 and 48 h post infection. Although fibroblasts are known to secrete low amounts of type I IFN, we were able to detect secreted IFNβ upon infection with HCMV WT (Figure 4B). Congruent with our transcript analysis, infection with HCMV UL35stop led to significantly higher levels of secreted IFNβ (Figure 4B). Altogether, we conclude that UL35 acts as a negative modulator of the IFNβ response during HCMV infection.

### 3.5. Stably Expressed UL35 Antagonizes PRR-Mediated IFNβ Transcription in HFF-1 Cells

Next, we generated HFF-1 cells that stably express C-terminally HA-tagged UL35 to analyze the IFN response upon PRR activation. UL35 expression was verified by immunoblotting (Figure 5A). Subsequently, we characterized the subcellular localization of UL35 by immunofluorescence with an HA-specific antibody. We observed a predominantly nuclear but also cytoplasmic localization of UL35 (Figure 5B), consistent with previous observations (Figure 3A,B). To examine the effect of UL35 on IFNβ transcription in HFF-1 cells, we activated the DNA-sensing cGAS-STING pathway by transfection of interferon stimulatory DNA (ISD) and measured IFNβ transcripts. In line with the previous reporter assays (Figure 1A), upon cGAS-STING signaling, IFNβ mRNA levels were significantly reduced in UL35-expressing HFF-1 cells as compared to control cells (Figure 5C). Next, we assessed the influence of UL35 on RIG-I signaling by infecting control or UL35-expressing HFF-1 cells with the RNA virus Sendai virus (SeV). In the presence of UL35, IFNβ mRNA transcript levels were significantly reduced (Figure 5D). In our previously performed ISG56 reporter assay, we have observed that UL35 did not interfere with IFNAR signaling (Figure 1F). To confirm this result in UL35-expressing HFF-1 cells, we stimulated the IFNAR pathway with recombinant IFNβ and measured mRNA transcript levels of two ISGs, IFIT3 and ISG56. We did not see a reduction of IFIT3 (Figure 5E) or ISG56 (Figure 5F) transcripts in the presence of UL35, confirming our previous observations and demonstrating that UL35 interferes with the type I IFN response upstream of the IFNAR pathway.

To determine the antagonistic effect of UL35 on PRR-signaling in a global manner, we performed whole cell proteomics using LC-MS/MS of poly(I:C)-stimulated HFF-1 expressing UL35 or empty vector control (Figure 5G, left panel). To exclude changes induced by ectopic UL35 expression itself, we also included non-stimulated cells. In total, 2,194 proteins were detected and enriched for gene Ontology (GO) terms (Appendix A). This GO term analysis allowed fast identification of factors known to be involved in IFN signaling. We identified several IFN-regulated proteins, including IFI16, IFIT3 and IFIT2, that were differentially expressed in UL35 expressing cells and control cells following poly(I:C) treatment (Figure 5G, right panel and Appendix A). In summary, we show that UL35 selectively targets IFNβ signaling downstream of multiple PRR in stably expressed primary HFF-1 cells, while IFNAR signaling is not affected by UL35.

### 3.6. The Interaction with O-Linked N-Acetylglucosamine (GlcNAc) Transferase Leads to GlcNAcylation of UL35, but Does Not Influence the Immune Regulating Function of UL35

O-linked N-acetylglucosamine (GlcNAc) transferase (OGT) has previously been identified as an interaction partner of transiently expressed UL35 and UL35a [22]. OGT catalyzes the posttranslational addition of GlcNAc to hydroxyl groups of serine and threonine residues of proteins. As OGT is involved in the modulation of PRR-mediated signaling by regulating NF-κB activation [40] and MAVS signaling [41], we wondered whether the interaction of UL35 with OGT contributes to UL35-mediated inhibition of the type I IFN response. First, we wanted to confirm the interaction of UL35 and UL35a with OGT by immunoprecipitation, and we included the UL35 homologue M35 in this analysis as well as MCMV M76. As a negative control, cells were transfected with an expression vector coding for beta-galactosidase (LacZ). As expected, endogenous OGT co-immunoprecipitated with myc-tagged UL35 and UL35a, while myc-tagged LacZ, M35, and M76 did not interact with OGT under our experimental conditions (Figure 6A).

To investigate whether OGT is involved in type I IFN modulation by UL35, we depleted OGT using an siRNA approach and conducted a cGAS-STING IFNβ luciferase reporter assay in HEK293T cells as described above. In control siRNA transfected cells, OGT expression was unaffected and as expected, IFNβ induction was reduced in the presence of UL35 (Figure 6B). Depletion of OGT resulted in a dramatic decrease of IFNβ reporter activity to similar levels in empty vector as well as UL35-transfected cells, which precluded a conclusion about the necessity of OGT for UL35-mediated inhibition of PRR signaling (Figure 6B).

Therefore, we assessed whether UL35 is post translationally modified by OGT, and whether this modification is important for the capacity of UL35 to inhibit PRR signaling. We used the YinOYang 1.2 server tool [42,43] to analyze the UL35 amino acid sequence for potential O-GlcNAcylation sites. We located predicted GlcNAcylation hotspots in the C-terminus of UL35, ranging from amino acid residue 500 to 550 (Figure 6C). To verify this prediction, we exchanged high-scoring serine and threonine residues for alanine by mutagenesis to prevent the addition of O-GlcNAc at these positions. In total, we generated four UL35 alanine mutants, which we designated Ala529-553, Ala529-531, Ala534/537, and Ala550-553 (Figure 6C). Next, we co-expressed untagged OGT together with HA-tagged versions of UL35 wildtype (WT) or the UL35 alanine mutants in HEK293T cells, immunoprecipitated UL35 with an HA antibody and performed immunoblotting for OGT as well as O-GlcNAcylation. As expected, OGT co-precipitated with UL35 WT (Figure 6D). Notably, we could detect a band corresponding to the size of UL35 with the O-GlcNAc-specific antibody, suggesting that UL35 is indeed modified by OGT. The UL35 Ala529-553 mutant showed neither interaction with OGT nor GlcNAcylation (Figure 6D). GlcNAcylation and OGT interaction of UL35 Ala534/537 and UL35 Ala550-553 were strongly reduced, whereas GlcNAcylation of the Ala529-531 mutant was comparable to UL35 WT, albeit it interacted only weakly with OGT (Figure 6D). These results suggest that the UL35 residues S534, T537, S550, S551, S552, and S553 are potential modification sites. To investigate the influence of GlcNAcylation of UL35 on its antagonistic function, we performed a cGAS-STING IFNβ luciferase assay with the UL35 mutants. We found that the UL35 mutants reduced IFNβ reporter activity similar as UL35 WT (Figure 6E). Therefore, we conclude that the interaction of OGT leads to GlcNAcylation of the C-terminus of UL35, but neither the interaction with OGT nor this modification influences the ability of UL35 to modulate the type I IFN response.

### 3.7. UL35 Antagonizes the Type I IFN Response Independent of the Cul4^DCAF1^E3 Ubiquitin Ligase Complex

Previously, the DDB1-Cul4 associated factor 1 (DCAF1) has been described as an additional interaction partner of UL35 [22]. DCAF1 is a known member of the Cul4^DCAF1^E3 ubiquitin ligase complex consisting of the scaffold protein Cullin4A (Cul4A), the E3 ligase Roc1, and the adaptor proteins DDB1 and DDA1. Because DCAF1 is utilized by multiple viral proteins to modulate the host [22,44,45], we asked whether this complex or the direct UL35 interaction partner DCAF1 itself contributes to the antagonistic function of UL35. Luciferase reporter assays of DCAF1− (Appendix A) and Cul4A− (Appendix A) depleted HEK293T cells showed that UL35 could still inhibit TBK1-mediated IFNβ activation. Therefore, we conclude that neither DCAF1 nor the E3 ubiquitin scaffold protein Cul4A is required for the IFN-antagonistic activity of UL35.

### 3.8. UL35 Interacts with TBK1 and Negatively Affects TBK1-Mediated Type I Interferon Activation

UL35 downmodulated the IFNβ response upon TBK1 activation (Figure 1D), but not upon expression of a constitutively active IRF3 (IRF3-5D) (Figure 1E). Therefore, we hypothesized that UL35 exerts its antagonistic function at the level of or downstream of TBK1 and upstream of IRF3. Notably, although the majority of tegument UL35 localizes to the nucleus during HCMV infection, a fraction of UL35 was detected outside the nucleus, which would be in line with cytoplasmic TBK1 or associated factors being its potential target. To compare signaling processes upon TBK1 and IRF3-5D stimulation in the presence or absence of UL35 in more detail, we co-transfected HEK293T cells with either empty vector (as unstimulated control), TBK1, or IRF3-5D together with UL35 or its empty vector control. Cells were lysed 20 h later and analyzed by immunoblotting for phosphorylated TBK1, phosphorylated IRF3, and total protein levels of both proteins. As expected, unstimulated control cells showed no activation of TBK1 or IRF3 (Figure 7A). In control cells, transfection of TBK1 led to strong phosphorylation of TBK1 and IRF3, indicative of a strong activation of innate signaling (Figure 7A). In contrast, the phosphorylation of TBK1 and IRF3 was drastically reduced by the presence of UL35 (Figure 7A). The phosphomimetic IRF3-5D mutant is a constitutively active form of IRF3 and therefore independent of activation events mediated by TBK1. Accordingly, endogenous TBK1 and IRF3 were not phosphorylated upon transfection of IRF3-5D, irrespective of the presence of UL35 (Figure 7A). Taken together, these data show that the presence of UL35 negatively affects the phosphorylation of TBK1 and subsequent downstream activation of IRF3.

Next, we asked whether UL35 interacts with TBK1 and thereby interferes with its activation. To analyze this, we co-transfected HEK293T cells with TBK1 and either UL35 or empty vector control and immunoprecipitated UL35 with an HA-specific antibody. Indeed, TBK1 specifically co-immunoprecipitated with UL35 (Figure 7B).

To verify the effect of UL35 on TBK1 phosphorylation in primary HFF-1 cells, we infected HFF-1 stably expressing empty vector or UL35 with HCMV UL35stop and lysed cells 0, 2, 4, and 6 h post infection and analyzed activation of the PRR signaling pathway by immunoblotting. In agreement with our previous findings (Figure 7A), the presence of UL35 reduced the phosphorylation of TBK1 as well as the phosphorylation of STING, IRF3, and STAT1 (Figure 7C). In addition, we transfected HFF-1 stably expressing ev or UL35 with ISD to stimulate the cGAS-STING pathway. Similar to the results with HCMV UL35stop infected cells, phosphorylation of TBK1, STING, IRF3, and STAT1 was reduced in the presence of UL35 (Figure 7D).

In summary, we conclude that UL35 associates with TBK1 to antagonize the activation of the TBK1-STING-IRF3 signaling platform in order to downmodulate the activation of type I IFN signaling.

## 4. Discussion

The ability of CMV to establish a life-long infection in its host has been attributed to its large coding capacity, which expresses a diverse range of immune evasion proteins. Viral manipulation of and escape from PRR-mediated detection of infection leads to a dampened antiviral type I IFN response. CMV tegument proteins are rapidly delivered into infected cells and several have been characterized to immediately exert their immune evasion activity [15,46,47,48,49,50,51,52,53].

Based on our previous finding that the MCMV tegument protein M35 inhibits the type I IFN response, we addressed whether its HCMV homologue UL35 may have a similar function during HCMV infection. Using quantitative luciferase reporter assays we could confirm that UL35 alone antagonizes IFNβ reporter activation upon PRR-stimulation. Similar to M35, UL35 inhibited cGAS-STING, RIG-I-MAVS, and TBK1-mediated activation of IFNβ transcription. In addition, we show that UL35, like M35, targets signaling upstream of the IFNAR. To our surprise though, UL35 did not inhibit activation of the IFNβ reporter following stimulation with the constitutively active IRF3-5D mutant. This is a clear difference to M35 and led us to conclude that UL35 and M35 follow different mechanisms to downmodulate PRR-mediated type I IFN induction.

By generating a UL35-deficient HCMV, UL35stop, and a mouse monoclonal antibody directed against the C-terminus of UL35, we were able to detect and verify expression and localization dynamics of endogenous UL35. When we followed the subcellular dynamics of UL35 during infection, we detected UL35 in the nucleus as early as 1.5 h upon infection. However, nuclear translocation was not complete as a smaller fraction of UL35 was still present in the cytoplasm throughout the infection. This observation raises the possibility that both the cytoplasm and the nucleus are potential sites of activity for UL35. When we compared the IFNβ response induced by HCMV WT and UL35stop in primary human fibroblasts, we could confirm the antagonistic function of UL35 in the context of infection.

To better understand the molecular mechanism of how UL35 modulates the type I IFN response, we first generated HFF-1 cells stably expressing UL35 and confirmed its antagonistic function upon PRR stimulation. To gain insights into the global proteome change upon PRR stimulation of UL35-expressing fibroblasts, we performed whole cell proteomics and could recover 2,194 proteins. We compared stimulated empty vector control cells to stimulated UL35-expressing cells and focused on proteins that are involved in IFN signaling. Several important proteins of the IFN signaling pathway were not recovered, most likely due to sensitivity limits. Nevertheless, we observed reduced expression of multiple proteins described to be induced by IFN signaling, such as IFI16, IFIT3 and IFIT2, in UL35 expressing cells. Interestingly, there was a significant reduction of the minichromosome maintenance (MCM) complex in UL35-expressing cells independent of poly(I:C) stimulation. The MCM complex is known to be an important factor for host DNA replication and modulates the cellular response to DNA double strand breaks [54,55]. Several HCMV proteins including IE2 [56] and pUL69 [57] target the host DNA replication machinery to force viral DNA replication. In addition, Qian and colleagues reported that the HCMV protein pUL117 inhibits the accumulation of MCM proteins to reduce cellular DNA synthesis [58]. Accordingly, our observation raises the question of whether UL35 contributes to the disruption of cellular DNA replication by downmodulating MCM proteins.

Salsman et al. (2012) previously used a mass spectrometry approach to identify cellular interaction partners of transiently transfected UL35 in HEK293T cells and described a role for UL35 in the DNA damage response [22]. DCAF1 was identified as a direct interaction partner of UL35 and shown to associate with UL35 nuclear bodies. In the same study, it was also reported that UL35 induces the accumulation of cells in the G2 cell cycle phase and activates the DNA damage and repair response in a DCAF1-dependent manner. To evaluate whether the interaction of UL35 with DCAF1 contributes to its inhibition of the IFNβ response, we performed an siRNA-mediated knockdown of DCAF1 and observed that UL35 could still inhibit IFNβ induction. In addition, we also depleted Cullin-4A (Cul4A), the scaffold protein of the Cul4A^DCAF1^ E3 ubiquitin ligase complex. Depletion of Cul4A also did not influence the inhibition of IFNβ transcription by UL35. Taken together, we conclude that the antagonistic function of UL35 on the IFN response is unrelated to its interaction with DCAF1.

We next assessed another reported interaction partner of UL35, the O-GlcNAc transferase (OGT), for its role in UL35-mediated IFNβ inhibition. OGT was identified as UL35 interaction partner by Salsman and colleagues [22]. OGT is a ubiquitous protein involved in many cellular processes including transcription, translation and stress responses [59]. Interestingly, the inhibition of OGT was reported to result in reduced HCMV titers [60]. OGT reversibly modifies serine and threonine residues of proteins with O-GlcNAc which is akin to phosphorylation events [61,62]. Recently, O-GlcNAcylation of MAVS was reported to be essential for the activation of type I IFN signaling in response to RNA viruses [41]. In addition, Yang et al. showed that O-GlcNAcylation of the NF-κB subunit p65 decreased the binding to IκBα and increased the transcriptional activity of NF-κB [40]. As OGT is progressively linked to host antiviral immunity, we investigated whether the interaction between UL35 and OGT was required to antagonize the IFNβ response. Knockdown of OGT resulted in a dramatic decrease of the type I IFN response in empty vector and UL35-expressing cells, confirming its crucial role for antiviral signaling, but prohibiting us from concluding that OGT is needed by UL35 to inhibit IFNβ activation.

We were, however, able to show that UL35 is GlcNAcylated between amino acid positions 534 to 553. To our knowledge, this is the second HCMV protein reported to be modified by OGT besides ppUL32 [63]. Moreover, the interaction with OGT and its GlcNAcylation was abolished by alanine substitution of serine and threonine residues of UL35 between amino acid position 529 and 553. As all UL35 alanine substitution mutants showed comparable inhibition of the IFNβ luciferase reporter, we concluded that OGT is not required for the antagonistic function of UL35. So why does UL35 interact with OGT? A possible scenario may be that UL35 sequesters OGT to enhance viral replication, since several cellular transcription factors involved in the regulation of the HCMV major immediate early promoter (MIEP) are modified by OGT [40,64,65]. Our mapping studies have opened up future paths to reveal the role of the UL35-OGT interaction during HCMV infection.

To this point, we have provided evidence that UL35 downmodulates the PRR-mediated IFNβ response downstream or at the level of TBK1 but upstream of the IFNAR, and we showed that DCAF1 and OGT do not contribute to its antagonistic effect. We further analyzed whole cell extracts of TBK1 transfected HEK293T cells and observed that in the presence of UL35, phosphorylation of TBK1 and IRF3 was remarkably reduced, mirroring the strong downmodulation in the TBK1-stimulated luciferase assay. Since IRF3-5D transfection stimulates IFNβ transcription independent of TBK1 activation or IRF3 phosphorylation, and HEK293T cells lack STING, these data suggested that UL35 may interact with TBK1 to interfere with IFNβ induction. Indeed, co-immunoprecipitation experiments showed that UL35 interacted with TBK1 in transiently transfected HEK293T cells. Subsequent experiments in fibroblasts stably expressing UL35 clearly showed that the presence of UL35 in HCMV-infected and ISD-stimulated fibroblasts led to a reduction of phosphorylated TBK1, STING, IRF3, and consequently, the downstream transcription factor STAT1.

Based on our findings we propose that the tegument protein UL35 is delivered into infected cells upon virion entry and binds to TBK1 in the cytoplasm, which acts downstream of both PRR cGAS and RIG-I, thereby interfering with activation of the STING-TBK1-IRF3 and MAVS-TBK1-IRF3 axis to antagonize IFNβ activation (Figure 8). While described functions of nuclear UL35 include the activation of the DNA damage response [22] or the activation of the HCMV MIEP [20], we have uncovered a novel role for tegument-derived cytoplasmic UL35. Up to now, several herpesviral proteins have been described to antagonize PRR-mediated signaling at the level of TBK1 [13]. For example, HSV-1-encoded UL46 (VP11/12) binds TBK1 and prevents TBK1 oligomerization [66]. Furthermore, HSV-1 ICP27 has been shown to bind to the activated STING-TBK1 signalosome to prevent IRF3 phosphorylation [26]. In addition, HSV-1 US11 binds to Hsp90 and promotes the proteasomal degradation of TBK1 [67]. Through this study, we have added to the multifunctional role UL35 plays during HCMV infection by identifying it as the first HCMV protein that associates to TBK1, thereby enriching our understanding of the many mechanisms HCMV employs to evade the immune response to establish a successful infection in the host.

## Figures and Tables

**Figure 1 microorganisms-08-00790-f001:**
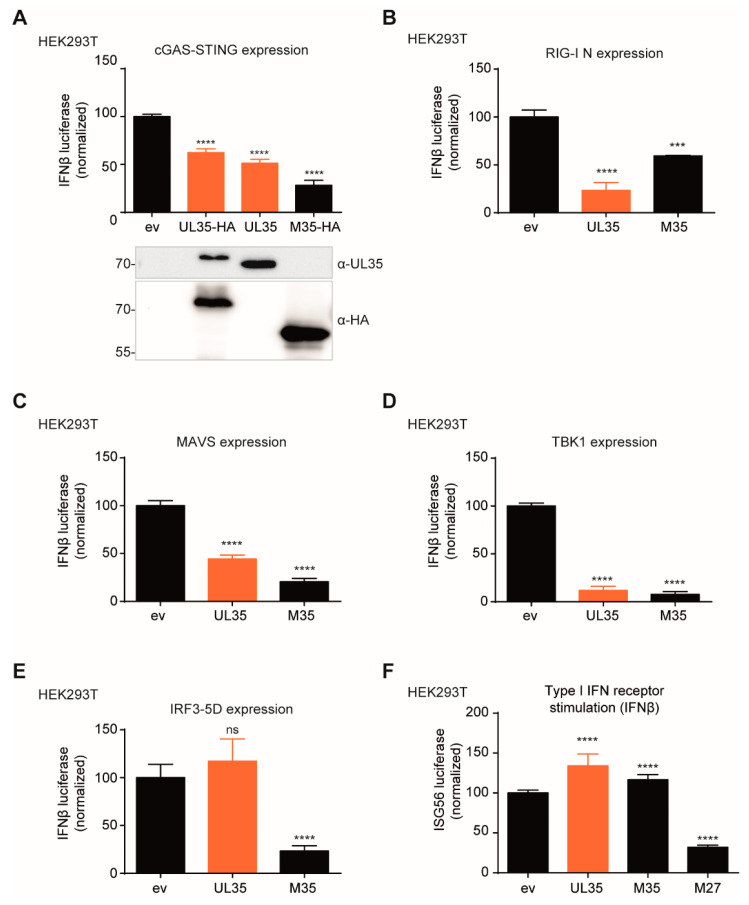
UL35 downmodulates transcription of IFNβ in luciferase reporter assays. (**A**) HEK293T cells were co-transfected with a reporter plasmid coding for firefly luciferase under the control of the murine IFNβ promoter (IFNβ-Luc) together with a Renilla luciferase normalization control (pRL-TK) and expression plasmids for IRES-GFP (unstimulated) or cGAS-GFP (stimulated), Cherry-STING and either empty vector (ev), HA-tagged M35, HA-tagged UL35 or untagged UL35. Cells were lysed 20 h later and luciferase activity was analyzed. An anti-HA antibody was used for detection of HA-tagged versions of UL35 and M35, and expression of untagged UL35 was verified with a UL35-specific antibody. (**B**–**E**) HEK293T cells were transfected with expression plasmids for IFNβ-Luc, pRL-TK and (**B**) RIG-I N, (**C**) MAVS, (**D**) TBK1, (**E**) IRF3-5D, or the respective ev. Cells were further co-transfected with either ev, HA-tagged UL35 or V5-tagged M35. Luciferase activity was analyzed as described in (**A**). (**F**) HEK293T cells were co-transfected with a reporter plasmid coding for firefly luciferase under the control of the human ISG56 promoter (ISG56-Luc), pRL-TK, and either ev, UL35-HA, M35-V5, or M27-V5. 24 h later, 0.1 ng/mL recombinant human IFNβ or vehicle only (unstimulated) was added. Cells were lysed 16 h post stimulation and luciferase activity was assessed. (**A**–**F**) Luciferase fold induction was calculated based on the firefly luciferase values normalized to Renilla from stimulated and unstimulated samples. Data was combined from at least three independent experiments and is shown as mean ± SD. Values were normalized to ev control. Significance compared to ev was calculated using the Student’s *t*-test (unpaired, two tailed), ns = not significant, *** *p* < 0.001, **** *p* < 0.0001.

**Figure 2 microorganisms-08-00790-f002:**
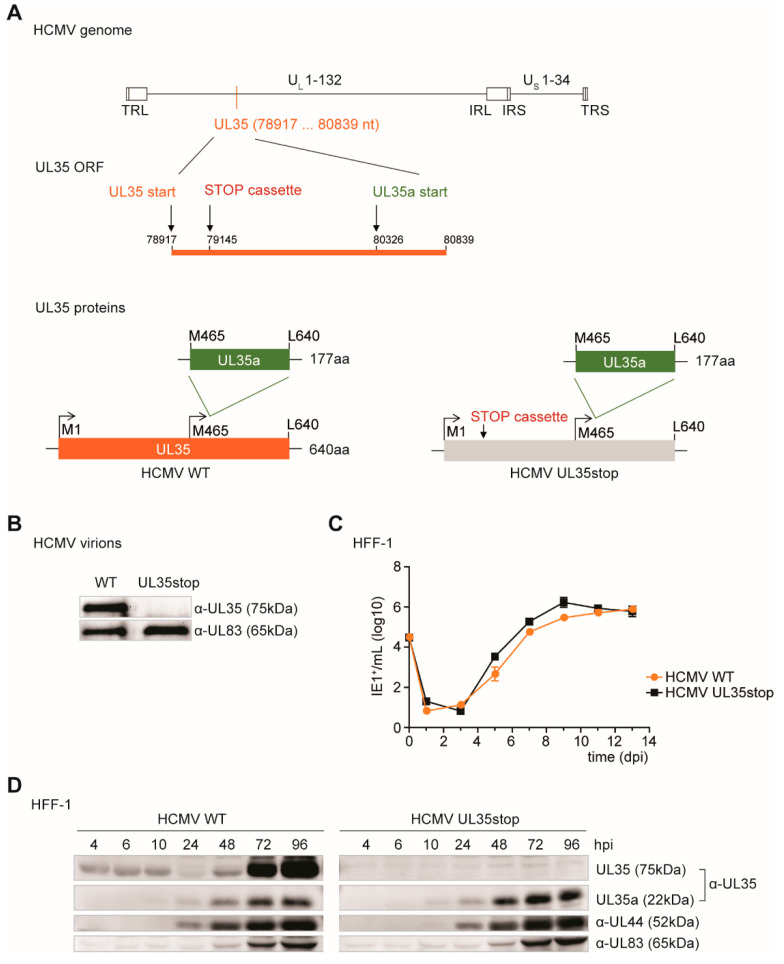
Construction of HCMV UL35stop by *en passant* mutagenesis. (**A**) Schematic representation of the recombinant HCMV UL35stop generated in this study. The location of the UL35 open reading frame is shown and position of the inserted STOP cassette is depicted. Numbers correspond to nucleotide locations in the genome (HCMV strain TB40/E, GenBank accession # EF999921.1). The scheme shows the translation products from the UL35 ORF of either wildtype (WT left panel) or UL35stop (right panel) HCMV. (**B**) Nycodenz-purified virus of HCMV WT and UL35stop was adjusted to 5 × 10^4^ viral particles based on the virus titer and subjected to SDS-PAGE. Immunoblotting was performed with antibodies specific for UL35 and UL83. (**C**) HFF-1 were infected with HCMV WT or UL35stop at an MOI of 0.1 and supernatant was harvested at the indicated time points. Fresh HFF-1 were infected with serial dilutions of the harvested supernatants for 48 h. Cells were then fixed with PFA and immunolabeled with an anti-IE1 antibody. IE1-positive cells were counted and the resulting titer (IE1^+^/mL) was log transformed to plot the growth curve. (**D**) HFF-1 were infected with either HCMV WT or UL35stop at an MOI of 0.1 and lysed at the indicated time points post infection. Whole cell lysates were analyzed by immunoblotting with antibodies specific for HCMV proteins UL35, UL44, and UL83. hpi: hours post infection, dpi: days post infection

**Figure 3 microorganisms-08-00790-f003:**
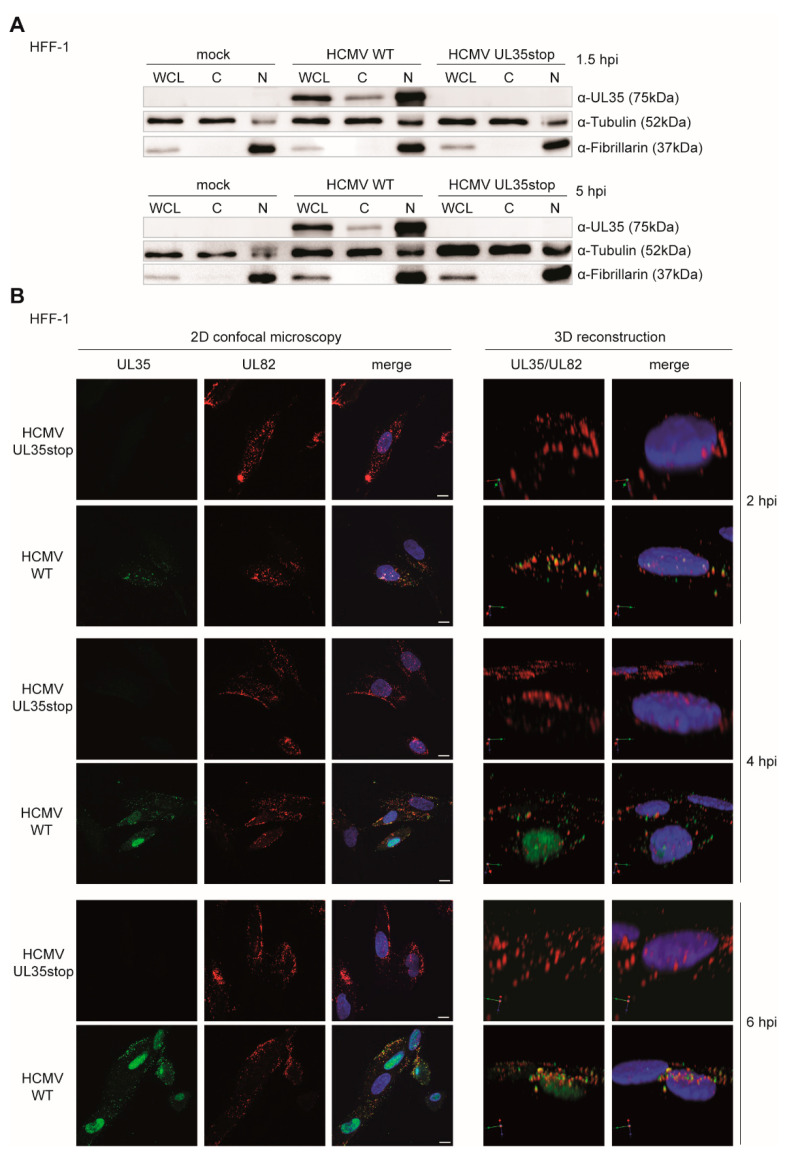
Dynamics of tegument-derived UL35 at early stages of HCMV infection. (**A**) HFF-1 were either mock infected or infected with HCMV WT or UL35stop for 1.5 h (upper panel) or 5 h (lower panel) at an MOI of 0.5. Whole cell lysates (WCL) were harvested and separated into cytoplasmic (C) and nuclear (N) fractions. Expression of UL35 was analyzed by immunoblotting with an anti-UL35 antibody. Tubulin and fibrillarin were used as controls for cytoplasmic and nuclear fractions, respectively. (**B**) HFF-1 were seeded on ibidi µ-slides and infected with HCMV WT or UL35stop at an MOI of 5. Cells were fixed at the indicated time points and immunolabeled with anti-UL35 (green) and anti-UL82 (red) antibodies. Nuclei (blue) were stained with DAPI. Images were acquired by confocal microscopy. Z-stacks were used for 3D reconstruction. Scale bars represent 10 µm. hpi: hours post infection.

**Figure 4 microorganisms-08-00790-f004:**
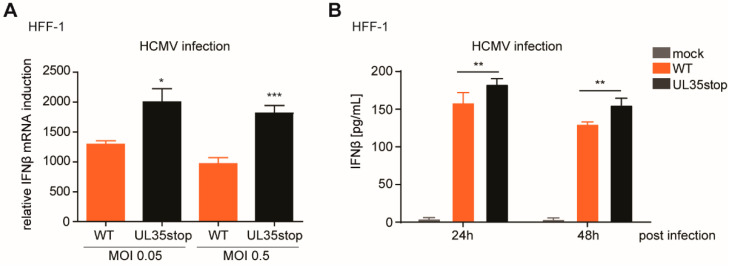
UL35 negatively modulates the type I IFN response upon HCMV infection. (**A**) HFF-1 were mock treated or infected with either HCMV WT or UL35stop at an MOI of 0.05 or 0.5. RNA was extracted 6 h post infection and the relative expression of IFNβ mRNA normalized to HPRT1 was quantified by RT-qPCR. (**B**) HFF-1 were mock infected or infected with either HCMV WT or UL35stop at an MOI of 0.5. Supernatants were harvested at the indicated time points and used for quantification of secreted IFNβ by MSD Multiplex assay. (**A**,**B**) Shown is combined data of three independent experiments as mean ± SD. Student’s *t*-test (unpaired, two tailed) was applied to compare HCMV WT and UL35stop. * *p* < 0.05, ** *p* < 0.01, *** *p* < 0.001.

**Figure 5 microorganisms-08-00790-f005:**
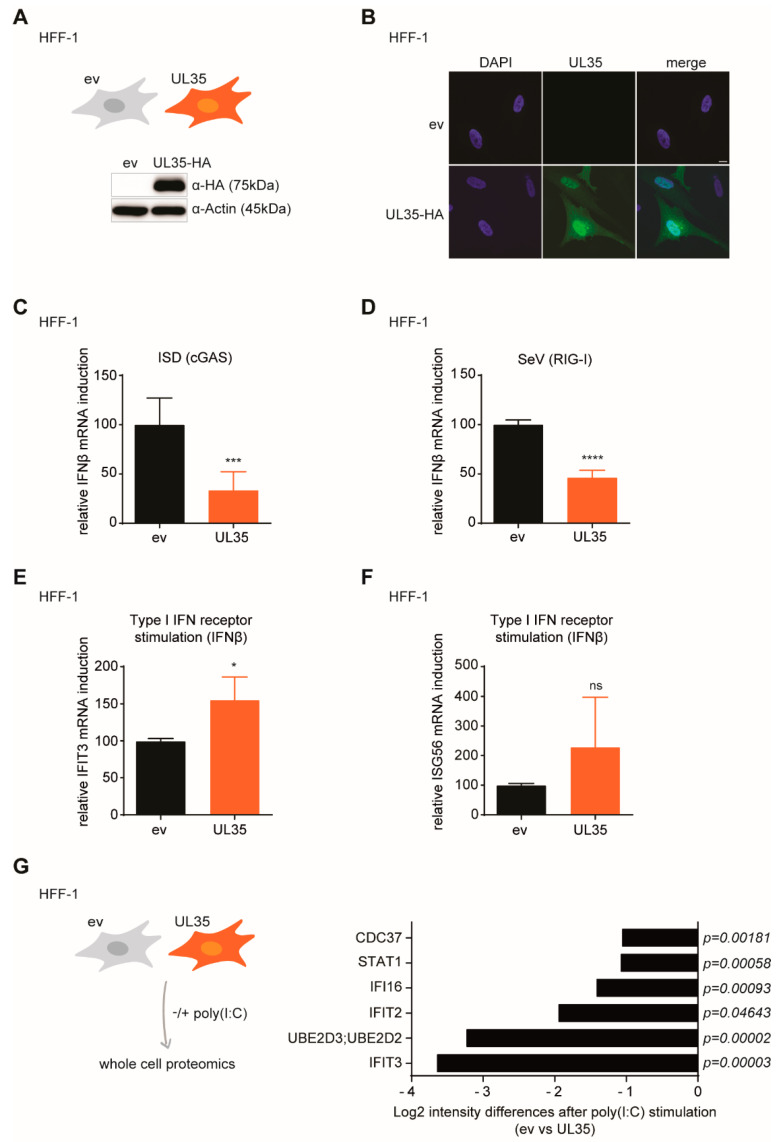
UL35 negatively modulates the IFNβ response in primary human fibroblasts. (**A**) Expression of HA-tagged UL35 was verified by immunoblotting of lysates from HFF-1 stably expressing UL35-HA or empty vector control. Actin served as loading control. (**B**) HFF-1 UL35-HA cells were analyzed by immunofluorescence with an HA-specific antibody (green) together with DAPI staining for nuclei (blue). Samples were imaged by confocal microscopy. Scale bar represents 10 µm. (**C**) HFF-1 stably expressing ev or UL35 were mock treated or stimulated by transfection with 2.5 µg/mL ISD. 4 h later, cells were lysed for RNA extraction and levels of IFNβ transcripts were determined by RT-qPCR. (**D**) HFF-1 stably expressing ev or UL35 were mock treated or infected with Sendai virus (SeV) for 6 h and IFNβ transcript levels were determined by RT-qPCR. (**E**,**F**) HFF-1 UL35-HA and control cells were stimulated with 20 ng/mL recombinant human IFNβ or mock treated. 6 h later, cells were lysed for RNA extraction and IFIT3 (**E**) and ISG56 mRNA levels (**F**) were analyzed by RT-qPCR. (**C**–**F**) HPRT1 was used as housekeeping gene. (**G**) HFF-1 stably expressing ev or UL35 were stimulated by transfection of poly(I:C) or medium only for 4 h. Whole cell lysates were prepared and peptides were subjected to LC-MS/MS for proteomic analysis (left panel). Proteome differences between stimulated ev− and UL35 expressing cells were compared and selected proteins are shown as Log2 (intensity) differences with the indicated *p*-values (right panel). Combined data of at least three (**C**,**D**,**G**) or two (**E**,**F**) independent experiments are shown. Student’s *t*-test (unpaired, two tailed) was applied to compare UL35 to ev, ns *p* > 0.05, * *p* < 0.05, *** *p* < 0.001, **** *p* < 0.0001.

**Figure 6 microorganisms-08-00790-f006:**
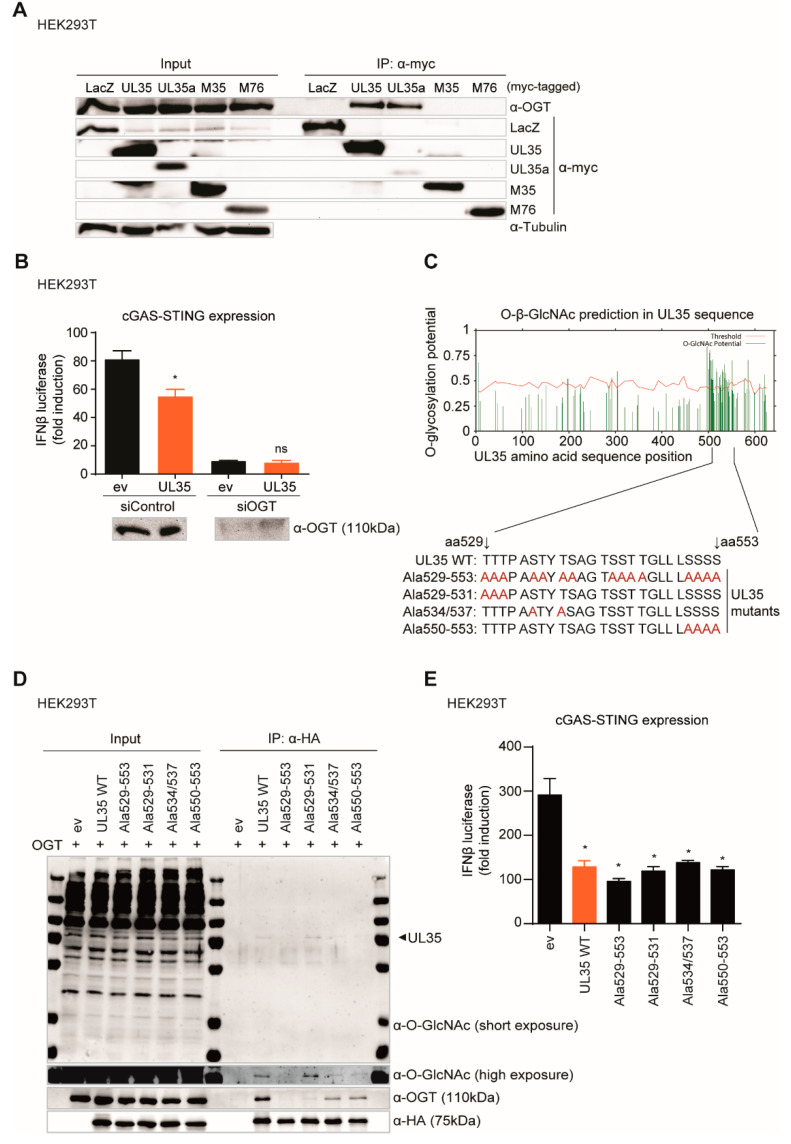
GlcNAcylation and OGT interaction of UL35 is not needed for its antagonistic function. (**A**) HEK293T cells were transfected with myc-tagged LacZ, UL35, UL35a, M35, or M76 and cell lysates were subjected to immunoprecipitation (IP) with an anti-myc antibody. Input and IP samples were analyzed by immunoblotting with antibodies specific for OGT, myc, and tubulin. (**B**) HEK293T cells were reverse transfected with either control siRNA or siRNA specific for OGT. 48 h later, cells were co-transfected with cGAS (stimulated) or IRES-GFP (unstimulated), Cherry-STING, IFNβ-Luc, pRL-TK, and either ev or HA-tagged UL35. Lysates for measuring luciferase activity were collected 20 h later and analyzed as described before. (**C**) Potential O-GlcNAcylation sites were predicted within the UL35 amino acid sequence (Ref. seq. YP_081494.1) using the YinOYang 1.2 online server tool. High potential serine and threonine residues within UL35 aa positions 529–553 were mutated to alanines (depicted in red). (**D**) HEK293T cells were co-transfected with expression constructs for OGT and either ev or HA-tagged UL35 expression constructs. 20 h later, proteins were immunoprecipitated using an HA-specific antibody and separated by SDS-PAGE. GlcNAcylation was analyzed by immunoblotting with an anti-O-GlcNAc antibody. OGT interaction was verified with anti-OGT antibody and UL35 expression with an anti-HA antibody. (**E**) HEK293T cells were co-transfected with cGAS (stimulated) or IRES-GFP (unstimulated), Cherry-STING, IFNβ-luc, pRL-TK, and either ev or HA-tagged UL35 WT or UL35 alanine mutants. 20 h later, cells were lysed and luciferase activity was measured. (**B**,**E**) One representative experiment out of three independent experiments is shown as mean ± SD. Student’s *t*-test (unpaired, two tailed), UL35 was compared to ev. ns *p* > 0.05, * *p* < 0.05.

**Figure 7 microorganisms-08-00790-f007:**
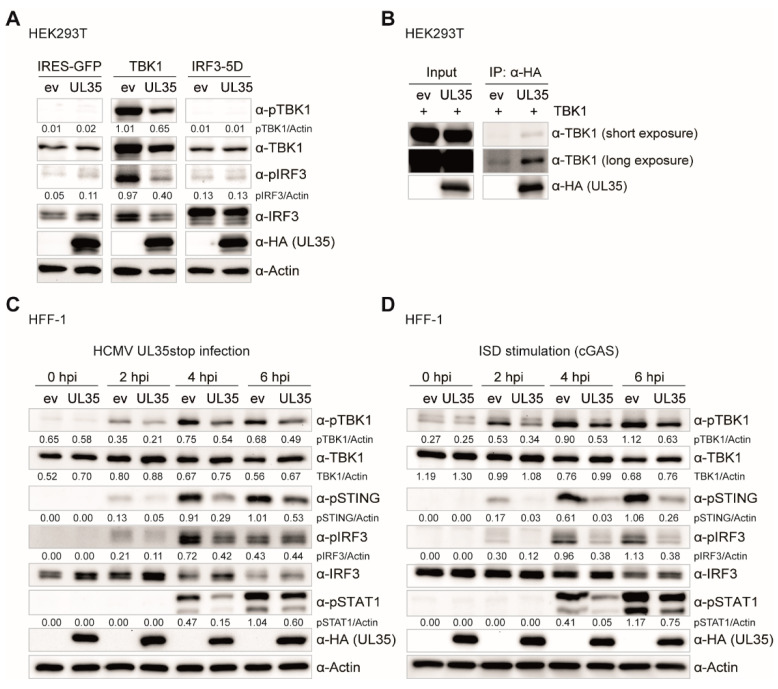
UL35 interacts with TBK1 and impairs the phosphorylation events within the TBK1-STING-IRF3 axis. (**A**) HEK293T cells were co-transfected with IRES-GFP (unstimulated), TBK1, or IRF3-5D together with either empty vector (ev) or UL35-HA. 20 h later, cells were lysed and whole cell lysates analyzed by immunoblotting for phosphorylated TBK1 (pTBK1), total TBK1, phosphorylated IRF3 (pIRF3), total IRF3, anti-HA for UL35, and Actin. (**B**) HEK293T cells were transfected with TBK1 together with either ev or UL35-HA. 20 h later, cells were lysed and 10% of the whole cell lysate was used for the input control. The remainder was subjected to immunoprecipitation with an anti-HA antibody to precipitate UL35. Input and IP fractions were analyzed by immunoblotting with anti-TBK1 and anti-HA antibodies. (**C**) HFF-1 stably expressing ev or UL35-HA were infected with HCMV UL35stop at an MOI of 0.1. Cells were lysed 0, 2, 4 and 6 h later and analyzed by immunoblotting with anti-pTBK1, anti-TBK1, anti-pSTING, anti-pIRF3, anti-IRF3, anti-pSTAT1, anti-HA, and anti-Actin antibodies. (**D**) HFF-1 stably expressing ev or UL35-HA were stimulated by transfection of ISD and immunoblotting was performed as described in (**C**). (**A**–**D**) One representative experiment out of two independent experiments is shown. Quantification of band intensities relative to the corresponding Actin band was performed using ImageJ and is shown below the corresponding immunoblot.

**Figure 8 microorganisms-08-00790-f008:**
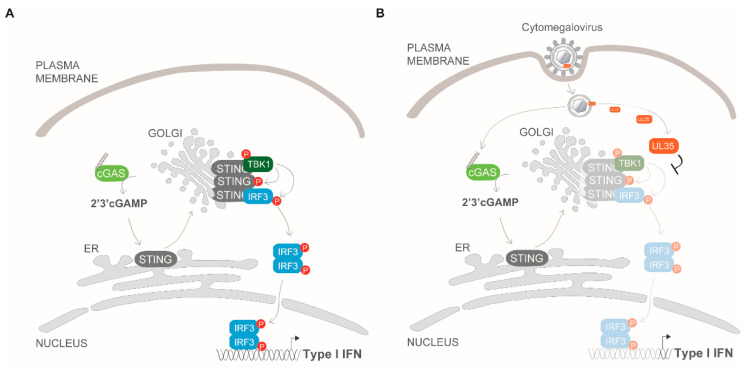
UL35 targets TBK1 to antagonize type I IFN signaling. Nucleic acids (**A**) or CMV infection (**B**) are recognized via sensors such as cGAS, which triggers signaling via TBK1 and IRF3 to induce transcription of IFNβ. (**B**) Following viral entry, tegument UL35 targets TBK1 to downmodulate the type I IFN response.

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
