# Peer review of "The Cytomegalovirus Tegument Protein UL35 Antagonizes Pattern Recognition Receptor-Mediated Type I IFN Transcription"

_microorganisms, 2020, doi:10.3390/microorganisms8060790_

Round 1

Reviewer 1 Report

Manuscript titled "The Cytomegalovirus tegument protein UL35 antagonizes pattern recognition receptor-mediated type I IFN transcription" is well written and well presented. There is sufficient background and experimental evidence provided to support the hypothesis and results. 

Reviewer 2 Report

Overal Assessment

This is a very well written manuscript that identifies a role for the human cytomegalovirus protein UL35 as a type I interferon antagonist. The authors hypothesize that the tegument protein UL35 from HCMV will have a similar function to the MCMV homologue M35 whose role as a type I IFN antagonist was previously published by this group. Although very similar to the previous publication in mouse I think that the low identity to the mouse homologue and the potential for a unique mechanism of IFNb antagonism merit publication of the current manuscript in the journal microorganisms.

Minor criticisms:

Introduction

Well written. No additional changes needed

Materials and Methods

Very detailed and sufficient to repeat all experiments. May be able to condense parts of this section with redundant use of some reagents and methods.

Results

3.1 - It is not clear how the authors came to the conclusion that UL35 antagonizes the type I IFN pathway through "multiple pattern recognition receptors". Multiple PRR's may use this pathway however multiple stimuli were not used for these experiments. Please clarify in this section.

Please justify the use of a mouse IFN beta reporter system when using a human cell line and a human virus.

In line 428 the authors state that the reduced IFNb transcription is reduced to "a similar extent". The figure suggests that the UL35-HA and the M35-HA are both lower but may be different when compared to each other. Was this verified by a statistical comparison? Please clarify by rewording or reporting statistic.

3.3 - The authors successfully demonstrate that the UL35 protein can be found in both the cytoplasmic and nuclear fraction from infected cell lysates and that this is lost with the UL35stop virus. Although this nicely demonstrates the feasibility of using the modified viruses, it only suggests that translocation to the nucleus is critical for blocking PRR signalling and does not determine where it is functioning. Determining how UL35 is trafficked to the nucleus would be helpful here and would provide a means to potentially inhibit this function. An expansion of this point in the discussion would be appropriate.

3.7 - The authors demonstrate here that UL35 associates with TBK1. I am not sure if this clearly demonstrates a direct interaction of TBK1 and UL35 as presented. I would like to see experiments performed with mutant TBK1 and mutant UL35 that demonstrate the precise binding residues responsible for driving this interaction. As this is presented I am only convinced that there is an association. Perform this or modify text in manuscript to more clearly state this conclusion in the results.

Reviewer 3 Report

The manuscript studied function of human cytomegalovirus (HCMV) tegument protein UL35 to determine the immune invasion mechanism by HCMV. Authors found that HCMV UL35 antagonize the type I IFN response at the level of TANK-binding kinase 1 (TBK1). It is an interesting study; however, the manuscript is too long, and results and discussion sections contain a lot of methodological descriptions. Duplicated descriptions of methodologies in these sections need to remove.

Comments

While HEK293T cells provide a convenient model to test the signaling mechanisms of type I IFN pathways against HCMV, the more complex regulation of this signaling pathway of IFNβ described in Figure 1 will require detailed analysis in more appropriate cell systems that have evolved to respond to HCMV with a high level of selectivity to determine immune invasion responses. Differences in the relative expression levels of MCMV M35, and HCMV UL35 may also account for some of the observed differences between mouse and human cells. It would be more relevant to compare HCMV M35 to UL35.

In figure 2, time points used in the growth curve and immunoblotting were not matching. In growth curve, cell growth of UL35stop showed decreasing at 14 dpi, whereas expression of both WT and UL35stop were increasing at 48, 72, and 96 dpi by immunoblotting.

Image of figure 3B is not clear. Different time points were used for immunoblotting and immunolabeling. Since immune image of UL82 was used and detected, subcellular localization of UL82 is also needed to analyze. The data between immunoblotting and imaging is not matching. In panel A, most of UL35 proteins is in nucleus at 1.5 hpi, but it is less obvious at 2 hpi in panel B and enrichment of UL35 were obvious in nucleus at 4 hpi and 6 hpi. Scale bars for z stacks are missing.

UL35 seems to interact with M35 in panels A and B of figure 6. Does this mean HCMV UL35 interacts with MCMV M35? Or myc antibody binds non-specifically to both M35 and UL35?

What is the rationale to use IRF3-5D instead of IRF3 alone? It is very confusing to understand section 3.8. Does IRF3-5D express IRF3 alone or IRF3 and 5? Downregulation of UL35 is not associate with activation of IRF3 (Fig 1E), whereas presence of UL35 is reduced the phosphorylation of TBK1 and IRF3 (line 697) even if the IRF3-5D is active mutant of IRF3, did not induce phosphorylation? How the active mutant form of IRF3 can be phosphorylated, irrespective of presence of UL35? In lines 709 to 713, description of figure 7C and D in text and figure legends are not matching.

Minor comments

Define PFA in line 226 and DAPI in line 393

In materials and methods, not all companies are supplied with their locations.

In line 332, detail about materials and instrument used to measure protein concentration in supernatant fraction are missing.

Specifics about the second antibody used for immunofluorescence is missing.
